# Long-Term Outcome after Liver Transplantation for Progressive Familial Intrahepatic Cholestasis

**DOI:** 10.3390/medicina57080854

**Published:** 2021-08-22

**Authors:** Safak Gül-Klein, Robert Öllinger, Moritz Schmelzle, Johann Pratschke, Wenzel Schöning

**Affiliations:** Deparment of Surgery, Campus Charité Mitte and Campus Virchow-Klinikum, Charité—Universitätsmedizin Berlin, 13353 Berlin, Germany; robert.oellinger@charite.de (R.Ö.); moritz.schmelzle@charite.de (M.S.); johann.pratschke@charite.de (J.P.); wenzel.schoening@charite.de (W.S.)

**Keywords:** liver transplantation, progressive familial intrahepatic cholestasis, long-term outcome

## Abstract

*Background and Objectives:* Progressive familial intrahepatic cholestasis (PFIC) is a rare autosomal recessive inherited disease divided into five types (PFIC 1-5). Characteristic for all types is early disease onset, which may result clinically in portal hypertension, fibrosis, cirrhosis, hepatocellular carcinoma (HCC), and extrahepatic manifestations. Liver transplantation (LT) is the only successful treatment approach. Our aim is to present the good long-term outcomes after liver transplantation for PFIC1, focusing on liver function as well as the occurrence of extrahepatic manifestation after liver transplantation. *Materials and Methods:* A total of seven pediatric patients with PFIC1 underwent liver transplantation between January 1999 and September 2019 at the Department of Surgery, Charité Campus Virchow Klinikum and Charité Campus Mitte of Charité-Universitätsmedizin Berlin. Long-term follow-up data were collected on all patients, specifically considering liver function and extrahepatic manifestations. *Results:* Seven (3.2%) recipients were found from a cohort of 219 pediatric patients. Two of the seven patients had multilocular HCC in cirrhosis. Disease recurrence or graft loss did not occur in any patient. Two patients (male, siblings) had persistently elevated liver parameters but showed excellent liver function. Patient and graft survival during long-term follow-up was 100%, and no severe extrahepatic manifestations requiring hospitalization or surgery occurred. We noted a low complication rate during long-term follow-up and excellent patient outcome. *Conclusions:* PFIC1 long-term follow-up after LT shows promising results for this rare disease. In particular, the clinical relevance of extrahepatic manifestations seems acceptable, and graft function seems to be barely affected. Further multicenter studies are needed to analyze the clinically inhomogeneous presentation and to better understand the courses after LT.

## 1. Introduction

Persistent neonatal cholestasis can be caused by a manifest liver disease [1,2]. Different anomalies of the liver and the bile duct tree, which can be acquired or hereditary, lead to disturbances of the bile flow. Progressive familial intrahepatic cholestasis (PFIC), an autosomal recessive inherited disease, induces impaired bile formation via mutated genes responsible for protein transporters. The altered hepatic bile flow subsequently leads, via severe intrahepatic cholestasis, to progressive chronic liver disease.

PFIC is divided into five types (PFIC1-5), while PFIC1 is the most common disease. The genetic causes of cholestasis assume an important role in understanding the physiology and pathophysiology of the liver. Type 1 (PFIC1) appears to be associated with defective bile acid secretion and is defined according to genetic criteria based on recessive mutations in the ATP8B1 gene on chromosome 18q21 [3,4,5,6,7,8]. The patients are characterized by jaundice and itching, and the subsequent progression of chronic liver disease [9,10,11]. Laboratory characteristics include glutamyl transpeptidase (GGT), whose activity is low for the degree of hyperbilirubinemia [12]. Bull et al. demonstrated good graft survival for PFIC 1 and PFIC2, consistent with earlier published LT series. Patients with PFIC2, which is based on a disorder of the bile salt excretion protein (BSEP) caused by a mutation of the ABCB11 gene, seem to present a slightly better survival rate [7,13]. The occurrence of malignancies for PFIC, especially that of HCC, is described in the literature [5,14,15,16,17]. Here, patients with missense mutations have been shown to be at lower risk for HCC [5,14,18].

Liver transplantation (LT) is an important therapeutic tool for the management of PFIC [19,20]. In the course of transplantation, a decrease in cholestasis has been observed, and the occurrence of PFIC1-specific extrahepatic manifestation was also described in the literature.

Extrahepatic manifestation post-LT as a potential complication, including pancreatic disease, is more frequently described after LT in PFIC1 [4,21,22,23,24]. Pancreatic disease may be associated with LT or be due to the progression of extrahepatic disease. In addition, diarrhea is also frequently described, as PFIC1 is strongly expressed in the small and large intestine of healthy individuals. Gastrointestinal dysfunction and pancreatic injury may contribute to diarrhea-associated malnutrition, poor growth, prolonged delay of puberty and low albumin and hemoglobin levels after LT in PFIC1 [4,23,25].

Histologically, steatosis may occur after LT and be associated with increased transminase activity, although graft function might not be compromised. The signal transmission between the hepatic and intestinal tract, which is disturbed in PFIC1 deficiency, seems to change after LT: post-transplant steatosis may be associated with such a change [4,26].

Follow-up of PFIC1 patients after LT is important for excellent clinical management and for anticipating results in the long term. Here, multicenter data analysis should provide more insight. The balancing act between the consequences of a persistent multisystemic disease with extrahepatic manifestations and graft dysfunction that may affect the LT outcome of recipients as well as graft survival and the need for LT must be addressed.

We here report long-term results of seven infants with PFIC1 who received LT as their first line of treatment before the age of two years. The clinical development of infants into adulthood was studied retrospectively, with special emphasis on the influence of PFIC1-typical extrahepatic manifestations and liver biopsies.

## 2. Materials and Methods

### 2.1. Study Population

A total of 219 patients younger than 18 years of age, who underwent LT from 1 January 2000 to 30 September 2019 at the Department of Surgery, Campus Charité Mitte and Campus Virchow-Klinikum at Charité-Universitätsmedizin Berlin, Germany, were retrospectively analyzed. The analysis and reporting of data received institutional review board approval (EA2/267/20).

Primary endpoints were recurrence of PFIC and/or graft-dysfunction. Secondary endpoints included 5-, 10-, 15-, and 18-year patient and graft survival as well as occurrence of extrahepatic manifestations.

Living liver donor transplantations were performed after acceptance by our institutional ethics committee and in accordance with standardized protocols. Liver transplants from deceased donors were organized by EUROTRANSPLANT within the framework of an allocation procedure.

For this purpose, demographic data, clinical features (physical development, histological examinations) and specific laboratory findings were collected at four different points in time (pre-transplantation, at the time of transplantation, short-term (first postoperative 90 days), and long-term (from the fourth postoperative month)) during the study period. This was based on annual check-ups and also on unplanned inpatient stays. Particular attention was paid to size–weight developments, the clinical course for different features (occurrence of splenomegaly, development of hearing disorders, complications though viral or bacterial infections), trough quantities of immunosuppressive drugs, associated coagulopathies and complications in general, as well as specific histological analysis of available liver biopsies.

Tumor follow-up was performed by means of regular ultrasound examination of the abdomen over the course of 6 months and AFP controls.

Standard immunosuppression initially was performed with tacrolimus ((tac); 0.1 mg/kg). Tac trough levels were kept at 10–15 ng/mL during the first month and afterwards at 5–7 ng/mL. Prednisolone tapered to 0.1 mg/kg after 10 days and was discontinued three months after LT. In the long-term follow-up, immunosuppression was extended to mycophenolate mofetil (MMF) in one patient and switched to MMF in four recipients. Patients were followed up until September 2019 for patient and graft survival.

### 2.2. Data Collection

Anonymous donor data were retrieved from the Eurotransplant Network Information System (ENIS), and additionally all available records of recipient data were collected from the hospital information system (SAP^®^ SE, Walldorf, Germany) and from archived patient files. Demographic and clinical characteristics of patients included gender, recipient and donor age at time of LT, PELD score prior to transplantation. In addition, previous abdominal and other disease-associated surgeries prior to LT, transplant types (living liver donation, deceased donor liver (split-graft, full-size organs), median surgery duration (min), and total hospital duration were also determined.

### 2.3. Statistical Analysis

Due to the limited sample size, the statistical analysis was performed in a purely descriptive manner without any testing. Calculations were performed using IBM SPSS Statistics, version 25 (IBM Corporation, Armonk, NY, USA).

## 3. Results

### 3.1. Characteristics Pre-Transplantation

A total of seven PFIC children out of 219 pediatric patients analyzed after LT were eligible for this analysis.

All infants suffered from PFIC1 (*n* = 7) and developed progressive cirrhosis (*n* = 7). The mean age at the time of LT was 9 months (range 7–23). All patients presented jaundice before liver transplantation, which regressed post-LT. In the long-term course, only one patient presented recurrent episodes of diarrhea, with no pancreatitis. No patient developed portal hypertension post-LT. Two patients had need of endoscopic retrograde cholangiopancreaticography (ERCP) due to biliary complications (biliary anastomosis stenosis and cholangitis). The patient with cholangitis showed no complications in the long-term follow-up after single ERCP and endoscopic papillotomy. The patient with biliary anastomosis stenosis showed a complication-free course after repeated ERCP and a Yamaka dilatation program over the last 12 years.

Among the seven patients, two pairs of siblings were detected (Table 1) and two patients developed multilocular HCC (Table 2). None of the patients had abdominal surgery prior to LT.

Different graft types were used for the LT, such as left-sided liver lobes from living donors (*n* = 3; mother, aunt, uncle), left-sided lobe split-grafts (*n* = 2) and full-size grafts (*n* = 2) from deceased donors (Table 3).

### 3.2. Short-Term Follow- and Long-Term Follow-Up

One patient developed acute rejection. One patient with known coagulopathy had a hematoma post-LT as a major complication according to Clavien–Dindo (>II), so that relaparotomy was required. No unplanned hospitalizations were detected during the short-term follow-up.

All patients showed a disease-free outcome (median; range 6 to 18 years) during the 17-year follow-up. Especially complications specific for LT, such as bile leakage or need for re-transplantation did not occur in any of the different donor types (LDLT, deceased organ transplantation, such as full-size and left-lateral split-grafts).

During this long observation period, the overall survival rate of all patients and grafts was 100%. None of the seven patients developed a PFIC1 relapse (Table 4). Both infants with multilocular HCC are disease-free up to now post-LT.

Patients showed varying degrees of extrahepatic manifestations, including intermittent mild diarrhea in one patient. Events of pancreatitis or persistent severe diarrhea were not detected in our patient cohort. In addition, hearing loss (*n* = 4) and growth failure (*n* = 2) were observed over the long-term course.

However, with regard to graft function, liver biopsies were performed in two male patients (siblings) with persistently elevated liver enzymes, resulting in steatohepatitis (Table 4). Histologically, steatohepatitis was described as mild and was associated with mild dysfunction due to slightly reduced levels of coagulation (INR) in one of the siblings (No. 4). Both HCC patients have demonstrated tumor-free follow-up. One patient with HCC received everolimus.

The consideration of general complications in long-term aftercare presented one patient (no. 2) with stenosis of the bile duct anastomosis and consecutive cholestasis 6.5 years after LT. Endoscopic interventions, such as repetitive dilatations (Yamakawa) were needed for one year. Currently, eleven years after the last endoscopic treatment, the patient is free of complaints and intervention. A second patient (no. 7) developed stenosis of the biliary anastomosis and of the bile duct, with consecutive cholangitis 6 years after LT, which was well-treated with endoscopic dilatation and papillotomy. Nevertheless, this patient received in consequence 10 years post-LT re-operation with biliodigestive anastomosis and is presently in good general condition. Another patient (no. 1) underwent partial lung resection 7 years after LT in recurrent pneumonia due to a middle lobe atelectasis.

## 4. Discussion

Progression of PFIC may temporarily present a clinical and laboratory improvement through drug therapies (rifampicin, ursodeoxycholic acid, cholestyramine, phenobarbital) or through biliary diversion (BD) in the absence of liver cirrhosis [27,28,29]. Even if there is no full manifestation of liver cirrhosis, liver transplantation (LT), especially for PFIC1, might also be the therapy of choice, since the PFIC-specific itching and jaundice can lead to serious developmental disorders, accompanied by retarded growth, severe rickets and reduction in quality of life [1,2,3,12,19,30,31,32,33,34]. LT for PFIC1 proves to be feasible regardless of donor type and reduces failure to thrive and permanent itching [19,30,35,36]. The analysis of our patient cohort showed an excellent course for all transplant types. Crucial remains the appropriate aftercare care depending on the underlying disease, regarding the different PFIC types [37,38,39,40].

We herein present long-term outcomes of 6 to 18 years for seven infants with PFIC1 undergoing LT at our surgical department. The seven infants (<2 years of age) in our analysis presented cirrhosis, with two patients exhibiting HCC (Table 2), and most were affected from birth with itching, pruritus and jaundice. At the timepoint of LT, the patients presented a median age of 9 months and had an inconspicuous course peri- and postoperatively.

Although the procedure of LT in infants has entered clinical routine with good results, regarding long-term follow-up, we must take into account late complications for patient- and graft-survival. In this context, for PFIC1, special attention must be given to extrahepatic manifestations occurring and exacerbating post-LT and the process of steatohepatitis [4].

Regarding extrahepatic manifestations, not only their occurrence but also their management play an important role in the routine follow-up. Since the gene for PFIC1 is mainly expressed in the intestine and pancreas, PFIC1 disease also plays a crucial role in extrahepatic tissues [8]. There are reports of severe clinical courses after LT [21,24]. Following this, PFIC1 patients after LT may develop persistent diarrhea and loose stools. This may explain a possible influence of PFIC1 on the intestinal absorption of bile acids, as patients with benign recurrent intrahepatic cholestasis (BRIC) have increased fecal bile acid loss and reduced bile acid reservoirs [41]. Furthermore, Pawlikowska et al. demonstrated in a multicenter analysis of 145 patients prior to surgical intervention in a comparative analysis between PFIC1 (*n* = 61) and BSEP (*n* = 84) that PFIC1 patients were more likely to have extrahepatic disease (16), whereas Walkowiak et al. showed that pancreatic secretion was normal in patients with PFIC1 both without and after LT for a prolonged period of time. They observed that steatorrhea was not associated with pancreatic insufficiency [22].

No pancreatitis was observed in our patient cohort. With regard to the often described diarrhea, only one patient in the retrospective study showed intermittent diarrhea, with no serious findings. Five out of seven children showed an age-appropriate development over the long term.

Apart from extrahepatic manifestations, liver function is of particular concern in long-term follow-up. Cuttilo et al. showed that LDLT with donor parents with a heterozygous PFIC1 status does not increase the risk of liver dysfunction in both the recipient and the donor [35]. Soubrane et al. described recurrence-free courses for 12 full organ LDLT [30]. However, the possible “recurrence of PFIC” remains a controversial issue. A possible alloimmunization after LT of the recipient against the BSEP, MDR3 (PFIC3) or FIC1 proteins of the liver donor remains a theoretical point of controversy. The possible susceptibility of PFIC patients to the FIC1, BSEP or MDR3 gene products due to a severe mutation is discussed. Davit-Spraul et al. discussed a rare case with a recurrence of pure hepatocellular cholestasis similar to PFIC2 in a male patient after cadaveric LT for PFIC2 [34].

Reports of long-term follow-up data with regard to histological findings of the graft and graft function as well are rare [3]. Only two patients out of the seven attracted attention with elevated liver enzymes. One of them presented mild steatohepatis.

A further point of our analysis is the first report—so far as we know—of the occurrence of HCC in two infants with PFIC1, and long-term results are provided, which are rather scarce [8]. However, our results are favorable, since both patients are disease-free.

Routine follow-up, in particular the closer and targeted oncological follow-up of HCC patients, includes the monitoring of liver function and the detection of extrahepatic manifestations. The anticipation of possible complications, based on the evaluations of long-term monitoring, is of particular importance. Especially from a socio-economic point of view, patients suffering from this rare multisystemic disease can maximally benefit from targeted aftercare during puberty into adulthood and be better integrated into social working life.

Certainly, our study is limited by its retrospective design and a small cohort. Nevertheless, LT for PFIC1 seems to be promising in our analysis in long-term outcome, because poor prognosis after LT in patients with PFIC1 was reported [26].

Biliary diversion can be successful if there is no fully developed cirrhosis of the liver and the disease is diagnosed early after the onset of symptoms in infancy [42].

Shiau et al. showed recently (Hepatology Communications) that an increase in Fibrosis-4 score was associated with 1.36-fold higher odds for LT in PFIC patients. In addition, the aminotransferase-to-platelet ratio index or Fibrosis-4 score appear to associate with risk for future LT in children with PFIC [43].

In our patient cohort, the full picture of advanced liver cirrhosis, associated with severe hypotrophy and developmental delay, was the decisive factor for the need for LT.

Our patients were symptomatic since birth and underwent LT at an age of 9 months old (median). Two of the patients transplanted at the ages of 18 and 20 months already had multiple HCC in cirrhosis. Therefore, after confirmation of the diagnosis and in the presence of advanced fibrosis or liver cirrhosis, liver transplantation should be sought as soon as possible.

As the urgency of transplantation is not adequately expressed by the labMELD for this group of patients, these patients should be assigned a matchMELD upon request. The disease should be listed under the criteria of the standard exceptions according to the German guideline. The risk of potentially developing HCC should be taken into account. In view of the often postpartum onset of the disease or early manifestation of symptoms, suspected HCC should be taken into account regardless of its size and extent, but after exclusion of distant metastases. As these patients are often infants, histological confirmation should not be mandatory, but should be supplemented by elevated AFP values, high-grade suspected imaging and the determination of biomarkers [43].

Finally, regarding the literature (Table 5) multicentric data with large patient cohorts, comparing PFIC patients of different subtypes after LT, are needed. An analog multi-center analysis for PFIC1 in comparison with PFIC2 might, facilitate the diagnosis and help to better differentiate the diseases.

The analysis and specific mapping of clinical courses and laboratory chemical characteristics in our PFIC1 patient cohort at four different points in time (before LT, at the time of LT, in the short and long-term follow-up after LT) may be beneficial in treatment before and after LT as well as for prediction of when transplantation is required.

Our aim was to identify characteristics that have been described so far and to analyze the patient and graft survival over time. In addition, a specific checklist, which comprehensively records associated symptoms, clinical expression, histological features and targeted controls in annual controls, could give us information about the prognosis in a multicenter evaluation [5,7,8,13,18].

## 5. Conclusions

Our results show that extrahepatic disease manifests only slightly, and that liver function is very good in the long-term follow-up. With a median follow-up of more than 14 years, only two patients developed fibrosis. The two patients with hepatocellular carcinoma continued to be free of recurrence during follow-up. Graft and patient survival in our cohort is excellent.

## Figures and Tables

**Table 1 medicina-57-00854-t001:** Patient characteristics and demographic features.

Pat.	Gender	* C/Affected Siblings	Premature Birth	Age: Onset of (Weeks)	Initial Symptoms **	Age at LT (Months)	** Indications for LT	HCC	Time of HCC Diagnosis	Follow-Up (Years)	∞ Outcome
1	f	no/no	no	since birth	j/c/p	8	LF	no	-	18	Disease-free
2	f	no/yes **	-	since birth	j/c/p	9	LF	no	-	18	Disease-free
3	m	no/yes **	yes	three months	j/c/p	20	SC, LF	yes	incidental	10	Disease-free
4	m	yes/yes **	-	since birth	j/c/p	7	LF	no	-	17	Disease-free
5	m	yes/yes **	yes	since birth	j/c	8	LF	no	-	14	Disease-free
6	m	no/no	no	since birth	j/c/p/d	18	HCC	yes	prior LT	6	Disease-free
7	w	no/no	no	since birth	j/c	23	LF	no	-	20	Disease-free

* Consanguinity: 2 and 3, and 4 and 5 are siblings; ** jaundice (j), pruritus (p), cholestasis (c), diarrhea; severe cholestasis (SC), liver failure (LF), HCC; ∞ outcome: including PFIC1 recurrence as well as HCC recurrence.

**Table 2 medicina-57-00854-t002:** HCC characteristics.

Pat.	Histological Type	Cirrhosis Type	Number of Nodules	Localisation of Nodules	Max. Diameter (mm)	N	V	G	AFP Pre-LT//Last Follow-Up
**1**	solid trabecular	micronodular	>3	bilobular	20	no	1	2	19,536//1.3
**2**	solid trabeculat	micro-, macronod.	3	bilobular	11	no	no	2	32,779//2.7

**Table 3 medicina-57-00854-t003:** Donor characteristics, perioperative parameters.

Pat.	Donor	Type	Operation Time (min)	∞ Re-Operation Post-LT	Other Perioperative Complications
1	Aunt	LDLT	245	no	no
2	Mother	LDLT	198	no	no
3	deceased	Full-size	289	no	no
4	deceased	split graft	242	no	no
5	deceased	split graft	349	yes *	no
6	Uncle	LDLT	340	yes **	no
8	deceased	Full-size	210	no	no

∞ Re-operation: within short-term, first 90 days post-LT; * reason: removal intraabdominal hematoma, familial APC-Resistency; ** calculated staged abdominal-wall closure.

**Table 4 medicina-57-00854-t004:** Histological findings and immunosuppression for short- and long-term follow-up.

	Short-Term Follow-Up	Long-Term Follow-Up
Liver-Biopsies #	2/7	4/7
Fibrosis	0/2	2/4
Cirrhosis	0/2	0/4
Steatohepatitis	0/2	3/4
ACR *	1/2	0/4
Tac	7/7	0/7
MMF	0/7	7/7
Liver function:		
Dysfunction **	0/7	1/7
Renal function:		
Dysfunction	0/7	0/7

# = in the short-term available for only two patients; in the long-term available for four patients. * ACR = histological proven acute cellular rejection; ** = elevated liver enzymes, hyperbilirubinemia, reduced INR.

**Table 5 medicina-57-00854-t005:** Review of the included literature specifically for PFIC1 patients after liver transplantation.

Authors (Reference)	Year	PFIC1 Specific	PFIC1 Patients	Age on Onset	LT	Graft Type	Time after LT	Survival	Outcome	Key Findings
Summerskill et al. [13]	1959	no	2	9, 17	no	-	-		-	Recurrent obstructive jaundice, itching; no histological findings
Ornvold et al. [14]	1989	yes	16	1–3 months	no	-	-	6 alive	n. m.	Classification: intrahepatic cholestasis without any structural abnormalities of the bile duct; death in early childhood; histological analysis revealing nonspecific cholestatic features
Soubrane et al. [31]	1990	yes	14	Neonatal period—5 years	yes	12 full-size grafts, 2 reduced-size grafts	17 months	yes	Uncomplicated	Question of recurrence remains, longer follow-up is necessary; medical therapy ineffective; LT good indication
Whiting et al. [33]	1994	yes	14	6.2 ± 3.6 months to years	yes	n. m.	n. m.	11 alive	n. m.	Only known effective therapy biliary diversion, LT; Better characterization of this syndrome is needed
Emond et al. [30]	1995	yes	11	4 years	yes	n. m.	3.8 years (mean)	8 alive	Uncomplicated	LT should be performed primarily in patients who have cirrhosis
Bull et al. [9]	1997	yes	9	11 weeks (mean)	no	-	-	n. m.	n. m.	Genetic analysis, light microscopy, transmission electron microscopy may help distinguish PFIC1 from others sorts of Bylers syndrome; Bylers syndrome is not a single clinicopathological entity
Jacquemin et al. [28]	1997	*yes*	9 out of 39 not exactly described	-//-	yes	n. m.	n. m.	n. m.	n. m.	In case of failure of UDCA treatment, partial external biliary diversion or liver transplantation
Arrese et al. [15]	1998	-	-	-	-	-	-	-	-	Molecular mechanisms of bile transport
Ismail et al. [29]	1999	yes	8 out of 46	10 months to 19 years	yes	6 cadaveric, 2 LDLT	n. m.	6 alive	Uncomplicated	Patients presenting cirrhosis or ineffective PEBD should qualify for LT
Jacquemin et al. [8]	1999	yes	-	-	-	-	-	-	-	PFIC is associated with mutations in hepatocellular transport system genes involved in bile formation
Knisley et al. [34]	2000	yes	-	-	-	-	-	-	-	In PFIC-1 LT improves life for the patient, who no longer is tormented by pruritus, but diarrhea, failure to grow and to enter puberty, and pancreatitis may persist as problems.
van Mil et al. [11]	2001	yes	-	-	-	-	-	-	-	It is clear that none of the speculative arguments described completely explain the etiology of FIC1 disease.
Egawa et al. [22]	2002	yes	7	3 to 18 years	yes	LDLT	1 year	7 alive	Remarkable improvements after LT all patients	Explanation for diarrhea and growth retardation after LT and importance of early treatment with bile adsorptive resin for postoperative diarrhea
Lykavieris et al. [6]	2003	yes	2	2.5 months; 2 months	yes	Cadaveric whole liver	9.5–11 years	2 alive	Complicated: diarrhea, liver steatosis, no catch-up of stature growth	PFIC1 characterized by normal GGT activity and extrahepatic features corresponds, which are not corrected or may be aggravated by LT, impairing QoL
Richter et al. [20]	2005	yes	1	2 years	yes	Cadaveric whole liver	2 years	alive	Uncomplicated	First report of PFIC and hepatoblastoma
Cutillo et al. [36]	2006	yes	7	27 months	yes	LDLT	n. m.	alive	Uncomplicated	Heterozygote familial donors can be safely used for LDLT, if evaluation of the donor does not reveal any previous history of even mild liver function defect
Knisley et al. [17]	2006	yes	11	13–52 months	yes	LDLT	n. m.	alive	Uncomplicated	PFIC associated with BSEP deficiency represents a previously unrecognized risk for HCC in young children
Özcay et al. [18]	2006	yes	2	n. m.	yes	LDLT	18.5 months (median)	alive	HCC recurrence free	LT may offer better survival rates
Pauli-Magnus et al. [2]	2006	yes	-	-	-	-	-	-	-	knowledge about the function of hepato-biliary uptake and efflux systems and discusses factors that might predispose to drug-induced cholestasis
Walkowiak et al. [23]	2006	yes	3	5–23 years	yes	n. m.	n. m.	n. m.	No observed pancreatic pathology; biopsy with normal intestinal mucosa	Pancreatic secretion was normal; observed steatorrhea not related to pancreatic insufficiency
Aydogdu et al. [26]	2007	yes	12	43.2 ± 27 months	yes	6 LDLT, 6 deceased donor grafts	1 year	8 alive	Good QoL, disappearance of pruritis, jaundice, improvement of nutritional status	LT is effective treatment with good outcome, minimal morbidity in PFIC patients with cirrhosis.
Miyagawa-Hayashino et al. [27]	2009	yes	11	4 years (median)	yes	LDLT	8.9 years (median)	8 alive	-	Recognition of post-LT steatohepatitis will provide a better understanding of the enterohepatic circulation of bile acids and the pathophysiology of intrahepatic cholestasis
Davit-Spraul et al. [10]	2010	yes	6	4 years (median)	yes	n. m.	n. m.	4 alive	Extrahepatic features were present after LT	PFIC1 patients present obvious signs of extrahepatic disease that do not resolve after LT.
Pawlikowska et al. [16]	2010	yes	61	66 months	yes	n. m.	n. m.	-	-	FIC1 patients showed stronger clinical and laboratory evidence of extrahepatic disease in comparision to PFIC2 patients
Hori et al. [37]	2011	yes	11	-	yes	LDLT	11.9 ± 4.5 years	8 alive	Digestive symptoms; steatosis and/or fibrosis were found; 3 dead patients had cirrhotic findings	LDLT should not be performed in PFIC1 patients until effective interventions can be made to correct the metabolic defects
Nicastro et al. [25]	2012	yes	1	5 months	yes	LDLT	30 months	alive	Complicated: Severe diarrhea, protein-losing enteropathy, steatofibrosis	EBD can avoid organ disfunction and loss in post-LT inpatients who develop graft steatohepatitis
Berumen et al. [24]	2014	yes	1	8 months	yes	Reduced size liver transplant	2 years	alive	Stricture of native bile duct; Persisting steatosis, severe diarrhea	Steatosis commonly occurs post-LT and can progress to fibrosis, cirrhosis with ned for re-transplantation
Mehl et al. [4]	2016	yes	-	-	-	-	-	-	-	Mutation of FIC1 protein significantly impairs bile salt secretion; exact mechanism for how FIC1 deficiency leads to cholestasis is not fully understood; varying severities of PFIC1 are noted
Liu et al. [3]	2018	yes	3	n. m.	yes	LDLT, DCD	1 year	all alive	Severe diarrhea	LT is currently the only effective treatment for end-stage liver diseases due to PFIC; LT for PFIC1 may worsen the extrahepatic manifestations
Wassmann et al. [42]	2018	yes	no specification for PFIC 1	-	-	-	-	-	-	Similar results HRQOL for PFIC patients after PEBD and LT; HRQOL in both groups even similar to that of healthy children. PEBD is primary surgical treatment for PFIC patients with cirrhosis
Baker et al. [32]	2019	yes		-	-	-	-	-	-	Clinical features and prognosis vary depending on the sub-type of PFIC For instance, outcomes after LT can be poor in patients with PFIC1, as the disease is multisystemic, and hence the bowel and lung manifestations are not necessarily improved post-LT

ICP: Intrahepatic cholestasis of pregnancy; PEBD: partial external biliary diversion.

## Data Availability

The data presented in this study are available on request from the corresponding author. The data are not publicly available due to the informed consent obtained from all subjects.

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
