# Peer review of "Long-Term Outcome after Liver Transplantation for Progressive Familial Intrahepatic Cholestasis"

_medicina, 2021, doi:10.3390/medicina57080854_

Round 1

Reviewer 1 Report

This paper presents an interesting experience of one center with seven PFIC1 patients who underwent liver transplantation, analyzing the short-term and long-term outcomes.

The manuscript needs some improvements:

  • The abstract should be rewritten as it misses the study's aim and needs to add more on methods. Also, the conclusions should be improved.
  • The Introduction must be improved, as the presentation of PFIC1 seems not well organized (for example, after a short introduction, liver transplantation is resented as a therapeutic tool, and then again, the types, clinical, and laboratory characteristics are presented.
  • The presentation of the results should be improved. Data from Tables 1 and 2 might be presented in the text as there is no need for the tables. The long-term weight and height presentation should also be made with a z-score to compare with the normal development of children of the same age. The Tabel 6 might be rearranged to have two columns for each characteristic presented (short term and long term).
  • The conclusions should be rewritten as there should consist of the study conclusions and not some general ideas or citations from other papers.
  • The use of the English language should be reviewed and corrected 
  • The references should be edited to fit the Journal requirements.

Author Response

Dear Reviewer 1,

Thank you for the review and we hope that we could answer your named points to your full satisfaction.

1. The manuscript needs some improvements: The abstract should be rewritten as it misses the study's aim and needs to add more on methods. Also, the conclusions should be improved.We thank you for the comment and have revised the abstract. In particular, we have worked out the aim and adjusted the methodology. In addition, we have rewritten the conclusions.

2. The Introduction must be improved, as the presentation of PFIC1 seems not well organized (for example, after a short introduction, liver transplantation is resented as a therapeutic tool, and then again, the types, clinical, and laboratory characteristics are presented.

We thank you for the good comment and have correctly restructured the introduction accordingly so that we first describe the PFIC types and only then move on to liver transplantation.

3. The presentation of the results should be improved. Data from Tables 1 and 2 might be presented in the text as there is no need for the tables. 

 We have adopted your suggestion to improve the presentation of the results and removed Tables 1 and 2. Accordingly, we have added an additional section under Results on page 3.
"All patients presented jaundice before liver transplantation, which regressed post-LT. In the long-term course, only one patient presented recurrent episodes of diarrhea, with no pancreatitis."
"No patient developed portal hypertension post-LT. Two patients had need of endoscopic retrograde cholangiopancreaticography (ERCP) due to biliary complications (biliary anastomosis stenosis and cholangitis). The patient with cholangitis showed no complications in the long-term follow-up after single ERCP and endoscopic papillotomy. The patient with biliary anastomosis stenosis showed a complication-free course after repeated ERCP and Yamaka dilatation program in the last 12 years."

4. The long-term weight and height presentation should also be made with a z-score to compare with the normal development of children of the same age. The Tabel 6 might be rearranged to have two columns for each characteristic presented (short term and long term).

We thank you for the suggestions and strongly believe that the manuscript has improved significantly as a result. Because the data on long-term weight and especially before liver transplantation are not available in a sustainably reproducible way, we have to refrain from presenting them with a z-score, to our regret. Nevertheless, we are convinced that the content of the article remains unaffected by this and ask for your understanding. We have rearranged the suggestions for improvement of Table 6 as desired and think that the table is now clearer in its message. Thank you very much!

5. The conclusions should be rewritten as there should consist of the study conclusions and not some general ideas or citations from other papers.

We thank very much for the request to revise the abstract. We have implemented this and adjusted the sources accordingly.

6. The use of the English language should be reviewed and corrected 

We have revised the English language again. 

7.    The references should be edited to fit the Journal requirements.

Thank you and we apologize for any inappropriate presentation. The references have now been edited. 

Yours sincerely,
Yours Safak Gül-Klein

Reviewer 2 Report

Congratulations on the excelent long term results in this subgroup of patients.  and nice literature review.

Authors suggested to compare results of liver transplantatioin children with PFIC1 or PFIC2. Did you transplanted only patients with PFIC1 in your center  or you did not include them in to the study intentionally?

Could authors explain why none of reported patients was treated  with non-transplant surgical treatment  before qualification for transplantation? Did all of them already present LF at diagnosis of PFIC?

As the paper retrospectively describes patients with PFIC1 transplanted several years ago authors did not discuss new approaches to the treatment of patients with PFIC and new pharmaceutical treatment trials which may lead to avoidance or delay in qualification for transplantation.  It seems that much less transplants were done in the last 10 years than in the period between 2000-2010 in patients with PFIC - is this reflection of other (better) diagnosis and approach for the treatment of infants with PFIC?

Author Response

Dear Mr. Reviewer 2,
Thank you for your comments on our manuscript.

1. Authors suggested to compare results of liver transplantation children with PFIC1 or PFIC2. Did you transplanted only patients with PFIC1 in your center  or you did not include them in to the study intentionally?

Thank you very much for your comment and specific question. We included all detected liver transplanted patients and accordingly did not intentionally exclude any patients. In the historical patient cohort consisting of 219 liver transplanted pediatric patients, we detected a total of seven children with Progressive Familial Intrahepatic Cholestasis (PFIC). Retrospective analysis of historical records from medical records throughout the treatment period showed that the children were classified as PFIC1 according to the best applied knowledge based on clinical presentation, laboratory parameters, and final histologic examination of the graft. 
It may be interesting in future studies to perform next gene sequencing of such historical cohorts and provide more insight into the PFIC types present. 

2. Could authors explain why none of reported patients was treated  with non-transplant surgical treatment  before qualification for transplantation? Did all of them already present LF at diagnosis of PFIC?

Thank you for the comment and your inquiry. We hope we can answer to your full satisfaction. Bringing in the aspect of subjecting patients to surgical treatment, like biliary diversion, before liver transplantation and associated possible improvement of clinical symptoms is correct and justified. 
The historical cohort covers a transplantation period from 1999 to 2014, and our retrospective research using available medical records allowed us to summarize that patients were treated conservatively as long as it was clinically justifiable. The persistence of failure to thrive and pruritus has led to an unstoppable and continuous deterioration of liver function, so that the children were planned by living liver donation in three cases and primary liver transplantation in four cases after availability of a suitable organ. This clinical approach is supported by consideration of final histology (4/7 cirrhosis, 3/7 hepatic fibrosis) as well as splenomegaly (4/7 twice each in the patients with cirrhosis and twice each in the patients with hepatic fibrosis). Therefore, it could be concluded that the patients were judged too poorly for non-trans related surgical treatment.

3. As the paper retrospectively describes patients with PFIC1 transplanted several years ago authors did not discuss new approaches to the treatment of patients with PFIC and new pharmaceutical treatment trials which may lead to avoidance or delay in qualification for transplantation.  It seems that much less transplants were done in the last 10 years than in the period between 2000-2010 in patients with PFIC - is this reflection of other (better) diagnosis and approach for the treatment of infants with PFIC?

We thank you for the relevant aspect that addresses future therapeutic approaches to this disease. 
Bosma et al (DOI: 10.3390/ijms22010273) reported in a review on Gene Therapy for Progressive Familial Intrahepatic Cholestasis that safety and efficacy studies are currently underway for Adeno Associated Viral gene therapy in adults.  Preclinical studies seem to suggest in the initial results that Adeno Associated Viral gene therapy may be applicable for PFIC.
The last transplant at our Surgical Hospital was performed in 2014. The remaining six children included two sibling pairs each. Therefore, we think this is most likely a realistic cross-section, with no relevant effect of newer therapeutic approaches on the current frequency of liver transplantation. This is supported by a similar paper by Henkel et al (DOI: 10.1111/petr.14108).

We thank you very much for the valuable review and hope that you find our manuscript suitable for publication.

Yours sincerely,
Safak Gül-Klein

Round 2

Reviewer 1 Report

The authors improved the manuscript according to my recommendations.

Still, some minor changes should be made:

  • in Table 4: there is no need for s-t and l-t as now there is no short form for these words; abbreviated words should be explained in the legend after the table
  • Table 4: liver function and renal function there is no need for two lines; choose which one you want to emphasize (normal or dysfunction) as the other is complementary.

Author Response

Dear Reviewer 1,

thank you again for your attention and the important note. We have now removed the unnecessary abbreviation in the heading of Table 4. In addition, we have changed the first row of the table from short-term and long-term to short-term follow-up and long-term follow-up. No more abbreviations occur in this regard.
Also, for liver and kidney function, we have indicated only dysfunction in each case, leaving only one row! 

Thank you from the bottom of our hearts for your review!

Kind regards,

Safak Gül-Klein